# Putative Wound Healing Induction Functions of Exosomes Isolated from IMMUNEPOTENT CRP

**DOI:** 10.3390/ijms24108971

**Published:** 2023-05-18

**Authors:** Paola Leonor García Coronado, Moisés Armides Franco Molina, Diana Ginette Zárate Triviño, Jorge Luis Menchaca Arredondo, Pablo Zapata Benavides, Cristina Rodriguez Padilla

**Affiliations:** 1Laboratorio de Inmunología y Virología, Facultad de Ciencias Biológicas, Universidad Autónoma de Nuevo León, San Nicolás de los Garza 66455, Nuevo León, Mexico; dianazt@gmail.com (D.G.Z.T.);; 2Facultad de Ciencias Físico Matemáticas, Universidad Autónoma de Nuevo León, San Nicolás de los Garza 66455, Nuevo León, Mexico; jorgeluismenchaca@gmail.com

**Keywords:** IMMUNEPOTENT CRP, dialyzable leukocytes extract, proteomics, wound healing, AKT

## Abstract

Chronic wounds in diabetic patients can take months or years to heal, representing a great cost for the healthcare sector and impacts on patients’ lifestyles. Therefore, new effective treatment alternatives are needed to accelerate the healing process. Exosomes are nanovesicles involved in the modulation of signaling pathways that can be produced by any cell and can exert functions similar to the cell of origin. For this reason, IMMUNEPOTENT CRP, which is a bovine spleen leukocyte extract, was analyzed to identify the proteins present and is proposed as a source of exosomes. The exosomes were isolated through ultracentrifugation and shape-size, characterized by atomic force microscopy. The protein content in IMMUNEPOTENT CRP was characterized by EV-trap coupled to liquid chromatography. The in silico analyses for biological pathways, tissue specificity, and transcription factor inducement were performed in GOrilla ontology, Panther ontology, Metascape, and Reactome. It was observed that IMMUNEPOTENT CRP contains diverse peptides. The peptide-containing exosomes had an average size of 60 nm, and exomeres of 30 nm. They had biological activity capable of modulating the wound healing process, through inflammation modulation and the activation of signaling pathways such as PIP3-AKT, as well as other pathways activated by FOXE genes related to specificity in the skin tissue.

## 1. Introduction

Exosomes (50–150 nm) produced by plants and animals (secreted from cells through an endosomal pathway dependent/independent of endosomal sorting complexes required for transport) [1] can induce diverse biological functions due to their content, such as DNA, mRNA, miRNA molecules, and proteins [2]. They possess characteristic biomarkers such as CD63, HSP90, HSP70, TSG1, and HMGB1 [3]. In addition to their function as delivery vehicles, these nanometric vesicles can act as components in cell communication as they can release bioactive compounds that possess stability, biocompatibility, and biorecognition [4]. They can be used as nano-vectors, as they can carry content able to induce signal pathway expression, similar to their parental cell origin [1,2,4]. 

Non-healing wounds greatly affect a patient’s quality of life, and represent an important economic burden. In the United States alone, they account for approximately USD 50 billion in healthcare costs each year, while scars from surgical incisions and trauma account for nearly USD 12 billion and burns account for USD 7.5 billion each year. Several therapies for wound healing are partially effective [5], but more effective therapies need to be developed.

Chronic wounds can take months or years to heal, and are characterized by high levels of inflammatory cytokines, loss of angiogenesis, and defective macrophages, etc. [6]. Peripheral neuropathy is one of the main causes of diabetic foot ulceration (DFUs) when peripheral nerves in limbs are involved [7]. Given that 85% of amputations are preceded by DFUs [7], they represent a health problem of importance. It has been reported that exosomes derived from mesenchymal stem cells used in a diabetic foot ulcer mice model induced accelerated wound healing through the activation of the PI3K-PTEN-AKT pathway [8,9]. In addition, adipose stem-cell-derived exosomes activated the same pathway in diabetic wounds [10].

IMMUNEPOTENT CRP (ICRP) is a mixture of low molecular weight substances that are less than 12 kDa, and is obtained from the dialyzable extract of bovine leukocytes from the spleen. It has been shown to be capable of modifying the immune response by regulating inflammation and inducing an antioxidant effect [11]. It has also been employed in clinical trials evaluating inflammation modulation after application in third molar extraction [12].

Splenic stem cells of Hox11 lineage can differentiate when injuries exist in pancreatic islet tissues [13]. Thus, the spleen is now also considered as a source for cellular replacement to ameliorate β-islet cells, suggesting the use of ICRP on diabetic wounds.

Based on the diversity of biological functions of ICRP, there have been research efforts to identify compounds responsible for each function. Nevertheless, there have not been conclusive results, except that ICRP has greater antitumoral activity when used as a complete spleen dialyzed extract. However, new biological applications can be elucidated more clearly when their peptides are identified and analyzed. Furthermore, previous studies using proteinase-K treatment on the whole extract increased antitumor and antioxidant activities in vitro [14]. Additionally, the high temperature (90 °C) required for its production [15] raises the suspicion that its activity could be attributed to the presence of exosomes. Therefore, we centered our analysis on this question. The main purpose of this paper is to determine the putative functions of ICRP using bioinformatic tools to analyze the proteomic content of the extract and exosomes isolated from it. These analyses can suggest the biological functions induced by the product and indicate the pathway of future investigations into diseases involving immunomodulation.

## 2. Results

ICRP contains exosomes capable of surviving the freeze-drying process of the commercial product. They maintain a semi-spherical shape of approximately 60 nm (Figure 1, Appendix A). Due to the nature of the AFM technique, it can be assumed that the exosome extraction method results in a pure sample, as can be seen in Figure 1. Charts with darker yellow coloration indicate a higher probability that certain processes would be induced. As observed in Figure 2 and Appendix A, the peptides present in ICRP corresponded to components characteristic of exosomes (rich in keratin filaments), with a *p* < 1 × 10^5^; moreover, they had catalytic activities, such as hydrolase and transferase activity, with the same *p*-value. Both the proteomic and AFM analyses confirmed the presence of exosomes exerting biological functions.

The analysis using GOrilla software identified that the outstanding biological processes induced by the ICRP were keratinization and cornification, in darker yellow (*p* < 1 × 10^7^ to 1 × 10^9^) in Figure 3, and the secondary processes included cellular response to chemokine, neutrophil degranulation, the regulation of the intracellular estrogen receptor signaling pathway, and the regulation of fat cell differentiation, in lighter yellow (*p* < 1 × 10^5^ to 1 × 10^7^) (Figure 3). Based on significant *p*-value, keratinization (4 × 10^6^) and cornification (5.98 × 10^6^) values are shown in Table 1. The former shows a higher enrichment value (1.61), based on the number of genes related to biological process (based on the mHG model with the FDR *q*-value (correction of the random error) in the correction of the *p*-value using the Benjamini–Hochberg logarithm, with only 2% chance of having false positives for the keratinization process and 2.25% for cornification (Appendix A).

These pathways coincide with the ones shown by Panther ontology (Fisher’s Exact test) (Figure 4, Table 2), especially inflammation mediated by chemokine and cytokine signaling pathway GO terms, with cellular responses to chemokines or neutrophil degranulation from GOrilla ontology GO terms, as well as between glycolysis GO term with the regulation of intracellular estrogen receptor signaling pathway GO term, as both programs show interactions between peptides present in the extract with components involved in those biological processes or pathways, not necessarily inducing them. It is worth mentioning that Panther ontology displayed results with significant FDR-*p* > 0.05 diminishing false positives, as observed in Figure 4b, where the biological pathway predicted less difference between “observed” and “expected” for inflammation mediated by chemokine and cytokine signaling pathways, validating the predictions made. 

The prediction of biological processes for exosomes based on the peptides present for a Homo sapiens model showed two main processes: neutrophil degranulation and cornification (*p* = 3.05 × 10^6^ for both) (Figure 5 and Table 3). Whilst the pathways showed a primary or better *p*-value (0.000004 for keratinization) in the ICRP extract, exosomes appeared to be secondary (0.0000178 for keratinization) but within same color scheme (Figure 1 and Figure 5). Nevertheless, one that remains in the strongest induced pathways for both is the cornification process (*p* = 0.00000598 for ICRP and *p* = 0.00000305 for exosomes). For exosomes, the pathways most strongly induced were granulocyte activation (*p* = 0.00000222), neutrophil activation (*p* = 0.00000222), myeloid leukocyte activation (*p* = 0.00000234), and cornification, as mentioned (Table 3). For *Mus musculus*, however, the catabolic proteasomal ubiquitin-independent protein prediction model emerged as the only pathway (*p* = 0.0000633) (Figure 6a), which coincides with keratin filaments for the *Homo sapiens* model both for exosomes and ICRP (Figure 2b and Figure 6b). Nevertheless, when introducing a target list in GOrilla ontology with peptides with an abundance ratio from 6.06% to 100% *p*-values change, keratinization (*p* = 0.00000738) and cornification (0.00000183) (Figure 7) processes move to the top of the list (Table 4). This demonstrates that the thousands of combinations made by the logarithm depend on the number of proteins/peptides in the target list, and so do the *p*- and *q*-FDR values. Therefore, keratinization cannot be ruled out as one of the possible main functions that exosomes could perform.

According to the Panther Classification System, peptides with an abundance of exosomes could interact with components of pathways relevant to wound healing, such as angiogenesis, the EGF receptor signaling pathway, the endothelin signaling pathway, the FGF signaling pathway, and the VEGF signaling pathway (Figure 8a), as well as B cell activation. When observing the biological processes (Figure 8b), exosomes play an important role in biological adhesion, which is correlated to a cell-to-cell junction; those peptides are also involved in biological regulation, localization, and signaling. The last process is the reason the next analysis was performed in the METASCAPE online server for the prediction of pathways, using a platform that also connects with drug target annotations and a protein atlas for tissue specificity. Through METASCAPE, we were able to observe a bar map in which color indicated a better *p*-value score for said process (Figure 9); those decisions were made taking into account the prioritization of genes. The program annotates enrichments, identifies statistically enriched pathways, and builds protein–protein interaction networks. Furthermore, enrichment in several genetic signatures such as cell types, transcription factors, and disease implications is carried out for quality control purposes. Terms with a *p*-value smaller than 0.01, a minimum count of 3, and an enrichment factor greater than 1.5 (the enrichment factor is the ratio of the observed counts to the counts expected by chance) are collected and grouped into groups based on their similarities of membership (subtrees with similarity greater than 0.3 are considered a group) and corrected using the Benjamini–Hochberg algorithm. This analysis coincides with some predictions made in GOrilla and Panther, such as regarding neutrophil degranulation and cellular response to stimulate angiogenesis at different levels (Figure 9 and Figure 10). As observed in Table 5, the process showed a better *p*-value score for neutrophil degranulation. Other biological processes of interest include, for example, salmonella infection; this does not mean that proteins support the infection, but exosomes contain peptides involved in the said process, which probably hijack infectious processes, coinciding with the immune response processes mentioned by the previous software (Panther 17.0). It is worth highlighting that the PID-ILK pathway mentioned in Table 5 as canonical under normal healthy conditions could be induced under exosome treatment in a diabetes context; the said pathway is involved in the cell-to-cell junction and extracellular matrix expression, both important for wound healing.

When peptides are grouped in a protein–protein network of interaction, niches are grouped per pathway, represented in Figure 11a by color, demonstrating that the majority of pathways have an intersection between one another. Results from GOrilla and Panther indicated that peptides have a catalytic effect, probably activating intermediates that have cross-talk between pathways. The majority of them were involved in the activation of the immune system, angiogenesis, and the development of the cornified envelope, among other things. In addition, in Figure 11b the protein–protein network interaction is shown with the intensity of the color based on *p*-value, locating the closest pathways and with the highest value to the center and right side of the scheme, this being the pathways’ cellular response to stimuli, supramolecular fiber organization, the regulation of the actin cytoskeleton, the regulation of the protein-containing complex assembly, translation, and the formation of a cornified envelope (*p* = 1 × 10^10^ to 1 × 10^20^). Moreover, peptides have more interaction in skin tissue (*p* = 1 × 10^15^) (Figure 12 and Appendix A) and seem to interact or induce the action of transcription factors such as SP1 and BRCA1, among others (Figure 13 by TRUST analysis *p* = 1 × 10^7^), with a remarque expression of FOXE1 target genes (*p* = 1 × 10^20^) (Figure 14).

For this reason, the PIP3-AKT signaling pathway was analyzed. In Figure 14, color boxes that represent pathway activation through the interaction between peptides belonging to exosomes with a component of the pathway can be observed, ending on the FOXO genes’ expression; these genes are involved in cell differentiation and migration, besides regulating inflammation. In Appendix A, interactors per component are shown considering the score from Reactome version 84 software as significant when bigger than 0.400. In addition to the PIP3-AKT pathway, other pathways seemed to activate when interaction occurred between peptides from exosomes, with protein components of the pathways’ cell–cell junction (Appendix A), signaling by FGFR1 (Appendix A), signaling by IGF1R (Appendix A), and signaling by Sonic Hedgehog protein (Appendix A) all being related to the wound healing process.

## 3. Discussion

Exosomes can be secreted by all cells in the body, and thus we hoped to observe their presence in a spleen extract such as IMMUNEPOTENT CRP, derived from a variety of cells that arrive in that organ. The presence of exosomes in IMMUNEPOTENT CRP was demonstrated by AFM and matches with the size range for an exosome, between 50 and 200 nm [1,16], showing an average size of 60 nm, in addition to the presence of exomeres around 30 nm, the last ones also participating in signaling pathways due to their protein and nucleic acid content [17,18]. Proteomic analysis demonstrated the presence of common exosome protein markers such as HSP90AA1, HSP90AB1, HSPA1A, HSPA2, TLN1, and HMGB1, characteristic of bovine exosomes [3]. Based on proteins present, the ontology enrichment analysis in GOrilla software for *Homo sapiens* predicted the intended final destiny of the product in the keratinizing and cornifying functions and the processes of cellular response to chemokines, neutrophil degranulation, the regulation of intracellular estrogen receptor, and the regulation of fat cell differentiation as secondary functions with statistical significance in processes, matching with the putative functions of the other program. Panther ontology for *Bos taurus* prediction also showed similar values for *Homo sapiens*, as observed in Figure 3. Focusing on the cellular response to chemokine and neutrophil degranulation could induce downstream signaling pathways, and within possible biological functions we could find angiogenesis and B cell activation, among others, that were observed in the Metascape program, allowing the possibility of treating infected wounds [19,20,21]. The formation of vessels and the control of possible infections are important for the wound healing process, whereas the keratinization and cornification process upstream functions could be induced by vessel formation in an infected wound, which could be reflected as a result, as was displayed in the Gorilla ontology of the most outstanding processes, resulting in a probable re-epithelialization of the wound. A remarkable result from Panther ontology that did not match with GOrilla ontology was the regulation over the cytoskeleton by GTPase Rho.

In the *D. melanogaster* cell repair processes, Rho has been reported to be required for the activation of myosin II, leading to its association with actin, which functions in the formation of actin filaments [22,23]. The ICRP has shown some processes to be secondary, whilst the same ones in exosomes stood out as primary processes; neutrophil degranulation correlated with the previous reports on the functions of IMMUNEPOTENT CRP, which modulates the immune system, probably due to exosomes activity [24]. However, the cornification process remains one of the most probable functions when examining proteins overexpressed in exosomes with respect to the whole extract (cornification process from IMMUNEPOTENT CRP extract, *p*-value = 0.00000598, and for exosomes *p*-value = 0.00000305), which means with higher enrichment in exosomes than in the whole extract. When the proteins overexpressed from 6.06% to 100% in exosomes were examined, the functions outstanding in the whole extract were the same for exosomes in a prediction model of *Homo sapiens* (keratinization *p*-value = 0.000004 and cornification *p*-value = 0.00000598 for IMMUNEPOTENT CRP, and keratinization *p*-value = 0.00000738 and cornification *p*-value = 0.00000183), increasing enrichment for the cornification process; this could imply that the biological effects exerted by the ICRP are attributed to the exosomes present in the extract, although a potency test is necessary to prove that. In addition, when cellular components of IMMUNEPOTENT CRP were evaluated, the presence of extracellular exosomes in the sample was highlighted with significant differences (sharing importance with keratin filament cellular components), for which the biological functions of the extract could be attributed to the exosomes, as those are one of the main components.

Moreover, the presence of specific types of keratins supports the hypothesis that the exosomes that possess keratin filaments present in IMMUNEPOTENT CRP can induce wound healing. In the exosomes, over-expressed keratins 1 and 2 can be found with respect to IMMUNEPOTENT CRP, and they have been reported to be overproduced in the healing process [25] in comparison with keratins 6 and 16, which are present in untreated skin; it is probable that the exosomes activate the cornification process by the presence of those keratin filaments.

In diabetic patients, wounds become chronic easily due to the lack of control of hyperglycemia, and these patients are reported to have defects in vessel formation, besides an inefficient closure of the wound with the formation of ulcers, and an inability to control infections [26]. The Panther ontology prediction of biological processes induced by exosomes indicated the activation of angiogenesis and VEGF signaling pathways, which are a key step in first phases of wound healing [26]. Furthermore, the expression of adrenaline and noradrenaline is implicated in the modulation of inflammation and expediting wound closure, besides the suggestion of protection against infection [27]. In terms of accelerating wound closure, biological functions of EGF and FGF signaling are probably induced, and neutrophil degranulation was remarked on if effective in Gorilla ontology; in Panther ontology, B cell activation was remarked on. Therefore, in a context of an infected diabetic foot ulcer, the exosomes could be employed as a treatment, supported in biological processes by Panther ontology, especially biological adhesion. Predictions in the METASCAPE online server remarked on possible pathways that match with previous software mentioned, and a new method of regulation for Salmonella infection that we hope can be used against infection, given that case reports have been published in human diabetic ulcers [28,29]. In addition, the heatmap from METASCAPE matched with the predictions from the last software, focusing on infection control, the formation of blood vessels, and the induction of proliferation in cells present in the dermis.

An Important concern is the site of action of a treatment; in our case, the main site of action was in skin tissue in a prediction for humans, suggesting a possible treatment employing exosomes isolated from IMMUNEPOTENT CRP in diabetic foot ulcers. We hypothesized that the exosomes activate proliferation-inflammation pathways, such as PIP3-AKT, as the pathway was significantly highlighted at different points, probably through the phosphorylation of MKRN1 and BAD, ending in the activation of genes; for example, FOXO genes correlating in Reactome and METASCAPE analysis. This pathway has been reported to induce wound healing in diabetic models [30], resulting in the differentiation of fibroblasts from myofibroblasts, which are the main cells in charge of wound healing in the first phases [31]. Other biological pathways highlighted were the cell–cell junction, the FGFR1 signaling pathway, the IGF1R signaling pathway, and the Sonic Hedgehog signaling pathway, all related to the wound healing process [32,33,34,35].

## 4. Materials and Methods

### 4.1. IMMUNEPOTENT CRP Protein Characterization

IMMUNEPOTENT CRP is a registered product composed of a mixture of substances of <12 kDa from spleen extract manufactured by LONGEVEDEN SA de CV, sold in a lyophilized presentation in Mexico. For proteomic analysis, a lyophilized pool was created from different batches to assure homology and representative data of the product. Then, this pool was analyzed using Tymora Analytics Operations (West Lafayette, IN, USA) to identify the proteins and digested to prepare a sample for liquid chromatography coupled to mass spectrometry coupled in tandem (LC-MS/MS), as described by Wu et al. [36]. The company sent us the list of proteins present in the extract, which was considered as the target list for bioinformatic analysis.

### 4.2. Exosome Proteomics Characterization

From the pool of ICRP sent to Tymora Analytical Operations Company (West Lafayette, IN, USA), a part of this mixture was separated to isolate the exosomes through EVtrap methodology with automated magnetic bead separation. After that, they were lysed to liberate protein content. Then, proteins were extracted and digested in an SDC buffer, and later proteins were identified through LC-MS/MS analysis as specified by Wu et al. [36]. This list of proteins was considered as a target list for bioinformatics analysis. Since background and target lists were available, the company provided the relative abundance (%) of each protein on the target list compared to the background list.

### 4.3. Exosome Shape and Size Characterization through Atomic Force Microscopy

Exosomes were isolated from a pool from different batches of lyophilized ICRP employing the kit Exoquick (Invitrogen, Waltham, MA, USA) that separates extracellular vesicles through ultracentrifugation. After that, the pellet was resuspended in sterile PBS 1X and analyzed with atomic force microscopy. The samples were observed using an NT-MDT Spectrum, NTEGRA Prima AFM at room temperature, with an RTESPA probe (Bruker, Billerica, MA, USA) of spring constant k = 40 N/m in intermittent contact mode. Images of height, deflection, and phase were obtained; 20 × 20, 10 × 10 and 5 × 5 µm^2^ image sizes were captured systematically for each sample at three different regions at least. The samples were analyzed with Gwyddion version 1.6 software to observe the morphological aspect of the exosome [16].

### 4.4. Biological Pathway Functions Prediction

The background and target lists were composed of the gene names that codified the proteins. First, putative biological pathway functions were predicted on the online software Gene Ontology enRIchment analysis and visuaLizAtion tool (GOrilla) using a background list and target list (selecting as prediction models *Homo sapiens* and *Mus musculus*). Two simulations were performed in the target list, one including all proteins present in the exosomes (1% to 100% relative abundance) and a second one with overexpressed proteins in exosomes (6.06% to 100% relative abundance) with respect to the background list; for predictions employing the target list, the background list was introduced as reference. The same procedure was performed in Panther ontology, specifying as model organisms *Bos indicus*, *Homo sapiens*, and *Mus musculus*. The software gave us data for molecular functions, cellular components, and biological processes.

For biological pathway prediction, proteins relating to transcription factors and tissue specificity in a *Homo sapiens* model, the Metascape v3.5.20230501software was used. In the software, the two lists function was used to introduce background and target lists with automatic parameters.

### 4.5. Evaluation of Wound-Healing-Related Signaling Pathways

Lastly, to identify the proteins (interactors) that interacted with components of cell signaling pathways related to wound healing in the literature and those correlated with the results obtained in previous software, Reactome version 84 software was used, entering background and target lists in the analyze gene list section (enabling the button to show interactors in the results), and with the PADOG analysis method, introducing “compare *Homo sapiens* with *Bos taurus*” in species comparison sections. The program showed yellow coloration in components of the pathway that show interactions with the proteins in the lists. With this information, Appendix A were built. Signaling component interactors present in exosomes and ICRP were identified, with their score of probability according to the Benjamini–Hochberg logarithm for multiple tests counting analysis.

## 5. Conclusions

We determined that the different biological activities of spleen cells’ dialyzed extract (IMMUNEPOTENT CRP) can in part be attributed to their total peptide content and peptides contained in exosomes, and not exclusively to a particular peptide. Furthermore, the in silico analysis coordinated with activities described before, such as anti-inflammatory capacity, but new activity, such as wound healing, was found, which could lead to new preclinical research to elucidate its activity, both regarding the effect of the total ICRP extract and its exosomes. Predictions indicated a putative route in which the ICRP could exert a wound healing process through inflammation modulation and the activation of signaling pathways such as PIP3-AKT, among others.

## Figures and Tables

**Figure 1 ijms-24-08971-f001:**
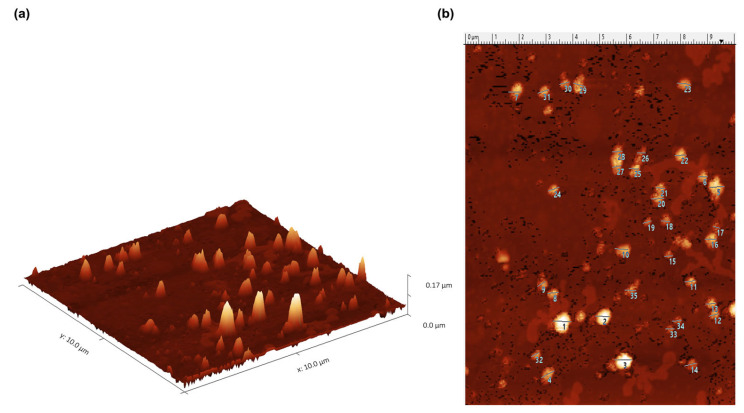
IMMUNEPOTENT CRP contains exosomes with a 60 nm average size. Exosomes were isolated with ExoQuick kit and characterized by AFM presenting a semi-spheroid shape. Exomeres of 30 nm were also observed. (**a**). Exosome distribution in 3 axes. (**b**). Two-dimensional distribution of exosomes (each exosome size is shown in Appendix A, listed by number).

**Figure 2 ijms-24-08971-f002:**
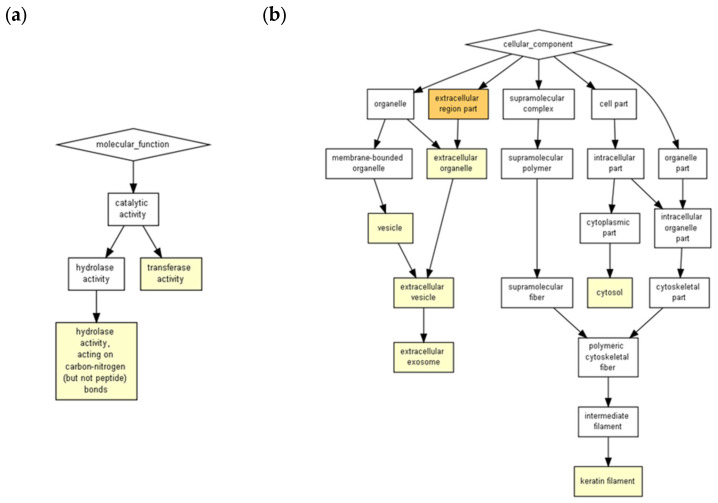
ICRP contains exosomes and peptides from the ICRP which putatively possess transferase activity and hydrolase activity acting on carbon-nitrogen (but not peptide bonds). The ICRP peptide content was entered into the GOrilla software with default values for predictions in Homo sapiens, considered significant when represented in a colored box for *p*-value < 1 × 10^3^ to *p*-value < 1 × 10^5^ and *p*-value < 1 × 10^5^ to *p*-value < 1 × 10^7^. (**a**) Exosomes possess catalytic properties such as hydrolase activity and transferase activity; (**b**) ICRP contains an extracellular component characteristic of exosomes and keratin filaments.

**Figure 3 ijms-24-08971-f003:**
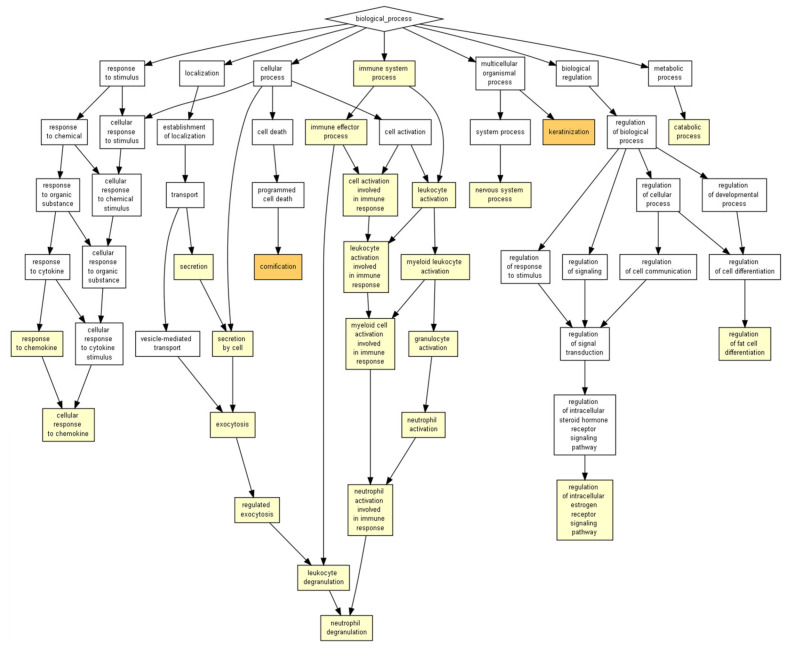
ICRP putatively and prominently shows keratinizing and cornifying functions as biological processes. The ICRP peptide content was entered into the GOrilla software with default values for predictions in *Homo sapiens*. Results considered significant are represented in a colored box when the *p* value < 1 × 10^3^.

**Figure 4 ijms-24-08971-f004:**
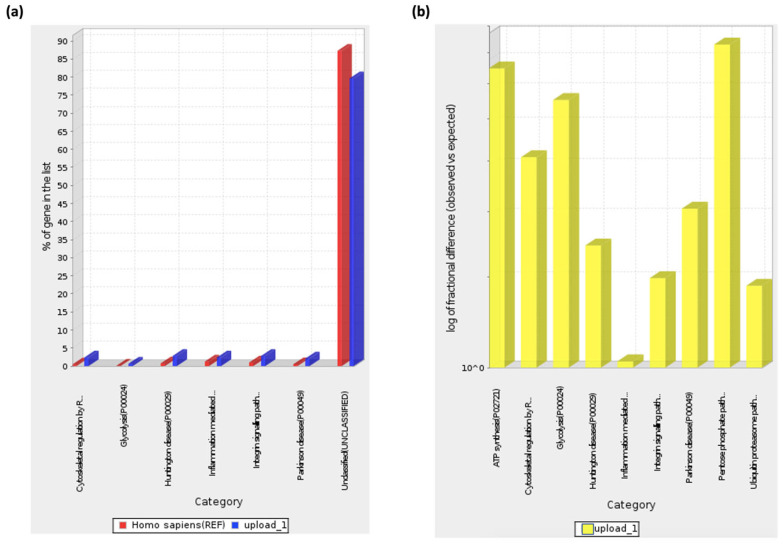
ICRP has peptides involved in the processes of inflammation mediated by cytokines and chemokines, Huntington’s disease, integrin signaling pathway, Parkinson’s disease, cytoskeleton regulation by GTPase Rho, and glycolysis. ICRP peptide content was entered into the Panther 17.0 ontology software with default values for predictions in *Homo sapiens* considering the *Bos taurus* proteomic origin. (**a**) Pathways induced based on gene percentage related to process (blue bars), in contrast with *Homo sapiens* reference lists per pathway (red bars). (**b**) Logarithmic fractional difference between observed and expected results.

**Figure 5 ijms-24-08971-f005:**
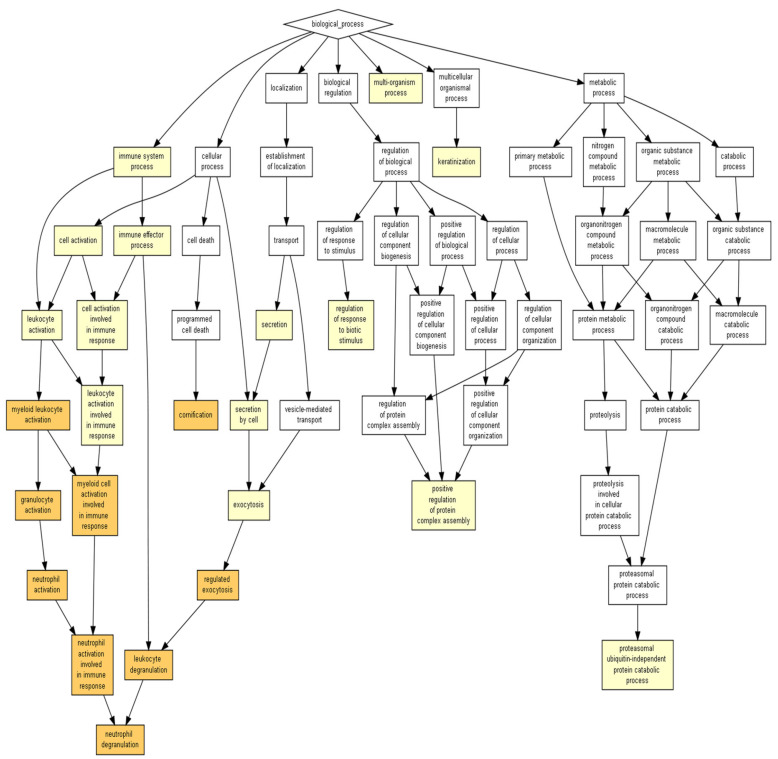
Exosomes possess neutrophil degranulation and cornification as suggested pathways. Based on proteins overexpressed in exosomes (target) relative to proteins present in the ICRP (background), 1–100% abundance ratio in a model of *Homo sapiens* (*p* < 1 × 10^5^).

**Figure 6 ijms-24-08971-f006:**
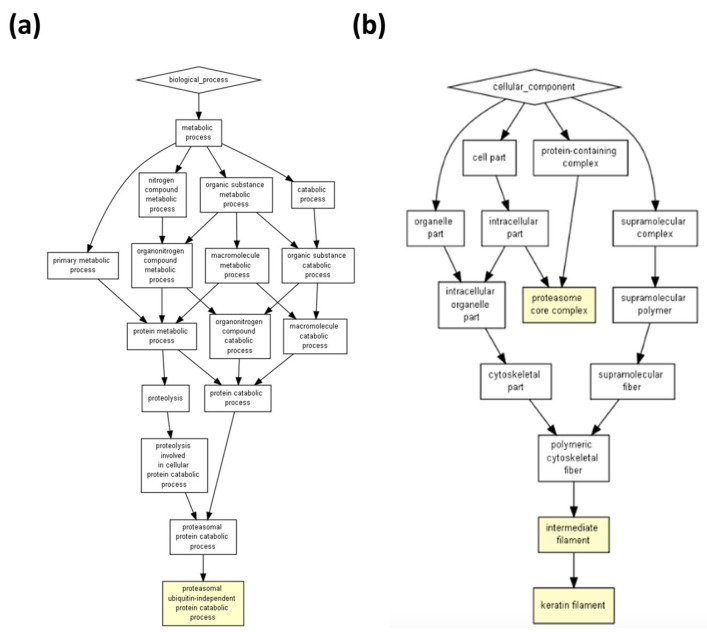
The proteasomal ubiquitin-independent protein catabolic process is suggested as a process pathway induced by ICRP-derived exosomes. Based on proteins overexpressed in exosomes (target) relative to proteins present in the ICRP (background), 1–100% abundance ratio in a model of Mus musculus (*p* < 1 × 10^3^ to *p* < 1 × 10^5^). (**a**) Shows proteasomal ubiquitin-independent protein catabolic as a biological process. (**b**) Shows keratin filament as an outstanding cellular component.

**Figure 7 ijms-24-08971-f007:**
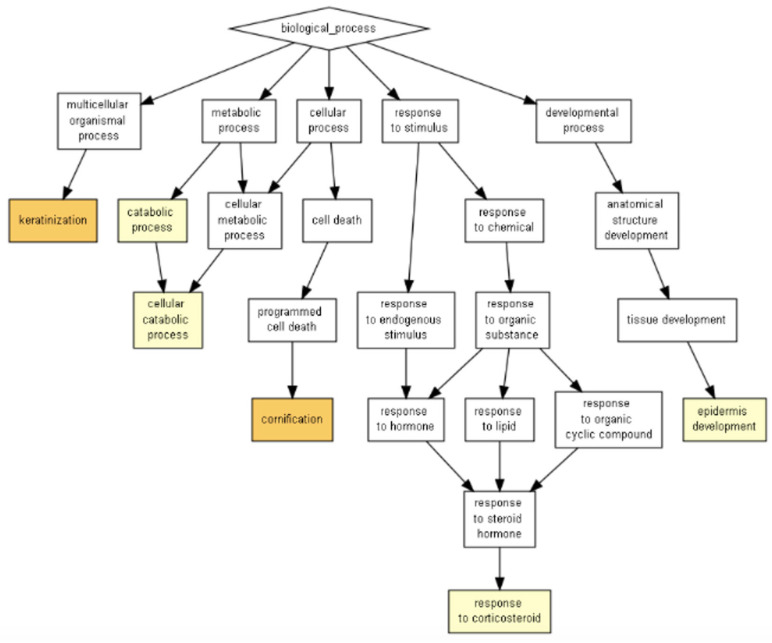
Cornification and keratinization as the main putative pathways induced by exosomes. Suggested functional pathways induced by ICRP-derived exosomes based on proteins overexpressed in exosomes (target) relative to proteins present in the ICRP (background), 6.06–100% overexpressed in a *Homo sapiens* model.

**Figure 8 ijms-24-08971-f008:**
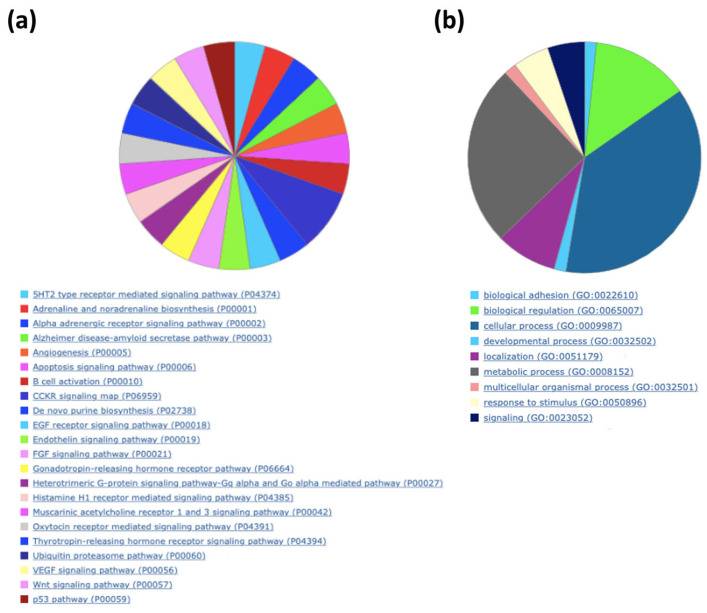
Peptides belonging to exosomes putatively induce Angiogenesis, B-cell activation, EGF receptor, FGF, and endothelin signaling pathways according to Panther Classification System. (**a**) Target list was analyzed (exosome peptides with abundance ratio from 1% to 100%) for a *Homo sapiens* model. (**b**) Biological Process (*p*-FDR < 0.05).

**Figure 9 ijms-24-08971-f009:**
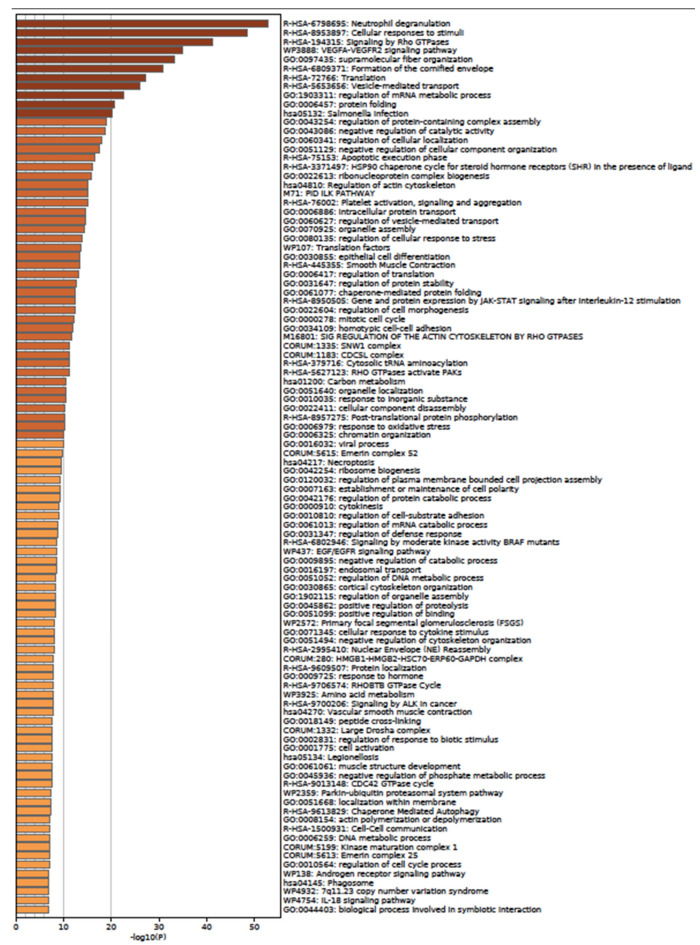
Peptides belonging to exosomes induce neutrophil degranulation, VEGFQQ VEGFR2 signaling pathway, regulation of cellular localization, HSP90 chaperone cycle for steroid hormone receptor, PID ILK pathway, epithelial cell differentiation, regulation of cell morphogenesis and EGF/EGFR signaling pathway. A background and target list (abundance peptide in exosomes from 1% to 100%) in METASCAPE online server and prediction for homo sapiens were performed with predetermined values.

**Figure 10 ijms-24-08971-f010:**
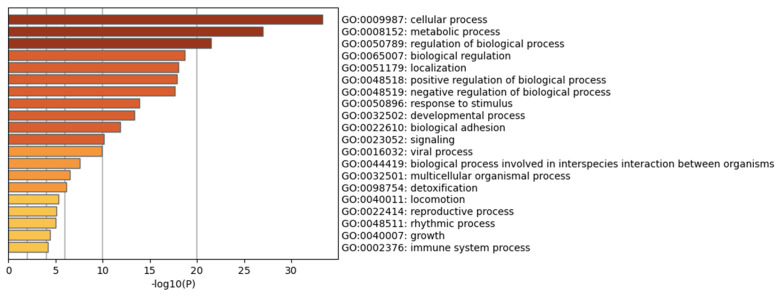
Peptides belonging to exosomes are involved in cellular processes, localization, signaling, and immune system processes. A background, and target list (abundance peptide in exosomes from 1% to 100%) in METASCAPE online server and prediction for homo sapiens were performed with predetermined values.

**Figure 11 ijms-24-08971-f011:**
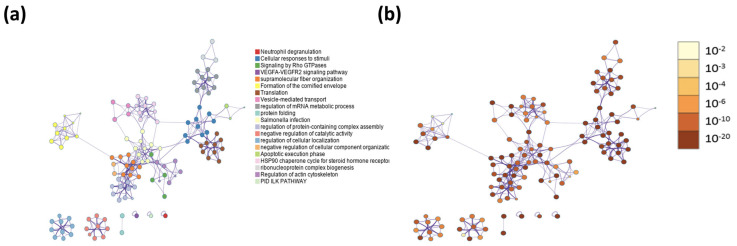
Pathways with better *p*-value scores are closely related and involved in cellular responses to stimuli, supramolecular fiber organization, and formation of cornified envelope. A background and target list (abundance peptide in exosomes from 1% to 100%) in METASCAPE online server and prediction for Homo sapiens were performed with predetermined values. (**a**) Protein–protein network interaction by pathway in colors. (**b**) Protein–protein network interaction by a pathway red intensity based on *p*-value.

**Figure 12 ijms-24-08971-f012:**
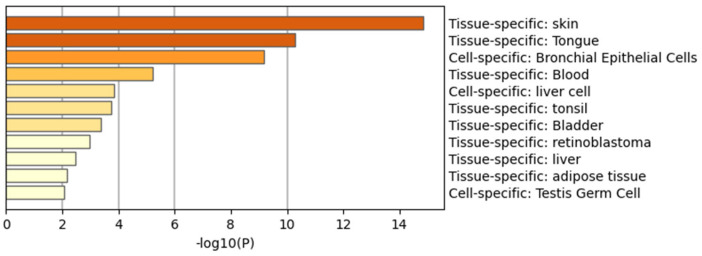
Peptides belonging to exosomes have an affinity to the skin. A background and target list (abundance peptide in exosomes from 1% to 100%) in METASCAPE online server and prediction for Homo sapiens were performed with predetermined values.

**Figure 13 ijms-24-08971-f013:**
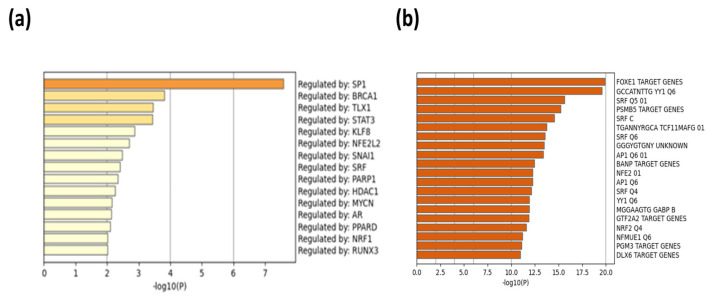
Transcription factors and target gene expressions are putatively induced by peptides belonging to exosomes. A background and target list (abundance peptide in exosomes from 1% to 100%) in METASCAPE online server and prediction for Homo sapiens were performed with predetermined values employing TRUST calculations. (**a**) The putative genes induced by the exosomes. (**b**) The transcription factor putatively induced by exosomes.

**Figure 14 ijms-24-08971-f014:**
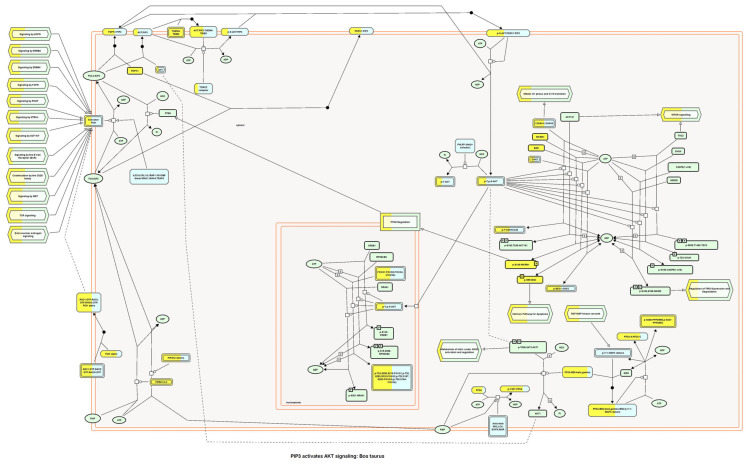
Exosomes derived from ICRP activate the PIP3-AKT pathway. A background and target list (abundance peptide in exosomes from 1% to 100%) in METASCAPE online server and prediction for Homo sapiens were performed with predetermined values corrected by the Benjamini– Hochberg algorithm.

**Table 1 ijms-24-08971-t001:** Biological processes induced by peptides present in the ICRP via GOrilla ontology.

Description	*p*-Value	FDR Q-Value	Enrichment (N, B, n, b) *
Keratinization	0.000004	0.0301	1.61 (1283, 40, 757, 38)
Cornification	0.00000598	0.0225	1.58 (1283, 44, 757, 41)
Exocytosis	0.0000369	0.0923	1.22 (1283, 160, 891, 136)
Leukocyte degranulation	0.0000521	0.098	1.25 (1283, 123, 891, 107)
Regulated exocytosis	0.0000541	0.0813	1.22 (1283, 154, 891, 131)
Immune effector process	0.0000609	0.0763	1.18 (1283, 169, 973, 151)
Granulocyte activation	0.0000674	0.0724	1.25 (1283, 122, 891, 106)
Neutrophil activation	0.0000674	0.0634	1.25 (1283, 122, 891, 106)
Myeloid cell activation involved in immune response	0.0000675	0.0564	1.25 (1283, 122, 891, 106)
Response to chemokine	0.000073	0.0549	28.51 (1283, 3, 45, 3)
Cellular response to chemokine	0.000073	0.0499	28.51 (1283, 3, 45, 3)
Neutrophil degranulation	0.0000846	0.053	1.25 (1283, 121, 891, 105)
Neutrophil activation involved in immune response	0.0000846	0.0489	1.25 (1283, 121, 891, 105)
Secretion by cell	0.0000888	0.0477	1.21 (1283, 173, 882, 144)
Myeloid leukocyte activation	0.0000952	0.0477	1.24 (1283, 125, 891, 108)
Leukocyte activation involved in immune response	0.000106	0.0499	1.20 (1283, 131, 973, 119)
Cell activation involved in immune response	0.000106	0.047	1.20 (1283, 131, 973, 119)
Leukocyte activation	0.000144	0.0603	1.18 (1283, 153, 973, 137)
Nervous system process	0.000164	0.065	2.77 (1283, 36, 232, 18)
Regulation of intracellular estrogen receptor Signaling pathway	0.00027	0.101	106.92 (1283, 4, 6, 2)
Secretion	0.000384	0.137	1.19 (1283, 178, 882, 146)

* Enrichment (N, B, n, b) is defined as follows: N—Total number of genes. B—Total number of genes associated with a specific GO term. n—Number of genes in the top of the user’s input list or in the target set when appropriate. b—Number of genes in the intersection. Enrichment = (b/n)/(B/N).

**Table 2 ijms-24-08971-t002:** Biological pathways induced by peptides present in the ICRP using Panther ontology.

PANTHER Pathways	#	#	Expected	Fold Enrichment	Raw *p* Value	FDR
Glycolysis	20	9	1.35	6.69	0.0000484	0.00129
Cytoskeletal regulation by Rho GTPase	86	33	5.79	5.7	2.24 × 10^13^	1.79 × 10^11^
Parkinson’s disease	101	29	6.79	4.27	1.82 × 10^9^	7.28 × 10^8^
Huntington’s disease	152	39	10.22	3.81	5.37 × 10^11^	2.87 × 10^9^
Integrin signaling pathway	200	39	13.45	2.9	4.26 × 10^8^	0.00000136
Inflammation mediated by chemokine and cytokine signaling pathways	261	33	17.56	1.88	0.00146	0.0334
Unclassified	17,971	1103	1,208.89	0.91	1.64 × 10^14^	2.62 × 10^12^

**Table 3 ijms-24-08971-t003:** Biological processes induced by peptides present in exosomes using GOrilla ontology (exosomes abundance ratio: 1–100%).

Description	*p*-Value	FDR *q*-Value	Enrichment (N, B, n, b)
Granulocyte activation	0.00000222	0.0167	1.32 (1283, 122, 764, 96)
Neutrophil activation	0.00000222	0.00835	1.32 (1283, 122, 764, 96)
Myeloid leukocyte activation	0.00000234	0.00587	1.32 (1283, 125, 764, 98)
Cornification	0.00000305	0.00573	1.53 (1283, 44, 764, 40)
Neutrophil degranulation	0.00000305	0.00459	1.32 (1283, 121, 764, 95)
Neutrophil activation involved in immune response	0.00000305	0.00383	1.32 (1283, 121, 764, 95)
Leukocyte degranulation	0.00000439	0.00472	1.31 (1283, 123, 764, 96)
Myeloid cell activation involved in immune response	0.00000599	0.00563	1.31 (1283, 122, 764, 95)
Regulated exocytosis	0.00000966	0.00807	1.26 (1283, 154, 764, 116)
Leukocyte activation involved in immune response	0.0000164	0.0123	1.28 (1283, 131, 764, 100)
Cell activation involved in immune response	0.0000164	0.0112	1.28 (1283, 131, 764, 100)
Keratinization	0.0000178	0.0111	1.51 (1283, 40, 764, 36)
Leukocyte activation	0.0000287	0.0166	1.25 (1283, 153, 764, 114)
Cell activation	0.0000751	0.0403	1.23 (1283, 161, 764, 118)
Immune effector process	0.0000887	0.0445	1.22 (1283, 169, 764, 123)
Multi-organism process	0.0000919	0.0432	1.19 (1283, 217, 764, 154)
Exocytosis	0.0000964	0.0426	1.23 (1283, 160, 764, 117)
Secretion by cell	0.000261	0.109	1.20 (1283, 173, 764, 124)
Secretion	0.000561	0.222	1.19 (1283, 178, 764, 126)
Proteasomal ubiquitin-independent protein catabolic process	0.000671	0.252	1.68 (1283, 14, 764, 14)
Regulation of response to biotic stimulus	0.000671	0.24	1.68 (1283, 14, 764, 14)
Positive regulation of protein complex assembly	0.000784	0.268	1.36 (1283, 52, 764, 42)
Immune system process	0.000933	0.305	1.14 (1283, 287, 764, 194)

**Table 4 ijms-24-08971-t004:** Biological processes induced by peptides present in exosomes via GOrilla ontology (exosomes abundance ratio: 6.06–100%).

Description	*p*-Value	FDR *q*-Value	Enrichment (N, B, n, b)
Cornification	0.00000183	0.0138	1.98 (1283, 44, 457, 31)
Keratinization	0.00000738	0.0277	1.97 (1283, 40, 457, 28)
Catabolic process	0.000202	0.507	1.30 (1283, 225, 457, 104)
Response to corticosteroid	0.000537	1	2.25 (1283, 15, 457, 12)
Cellular catabolic process	0.000566	0.851	1.28 (1283, 212, 457, 97)
Epidermis development	0.000945	1	2.34 (1283, 12, 457, 10)

**Table 5 ijms-24-08971-t005:** Pathways induced by exosomes (abundance ratio 1–100%) according to METASCAPE.

GO	Category	Description	Count	%	Log10(P)	Log10(q)
R-HSA-6798695	Reactome Gene Sets	Neutrophil degranulation	94	11.75	−52.85	−48.5
R-HSA-8953897	Reactome Gene Sets	Cellular responses to stimuli	111	13.88	−48.49	−44.45
R-HSA-194315	Reactome Gene Sets	Signaling by Rho GTPases	98	12.25	−41.21	−37.59
WP3888	WikiPathways	VEGFA-VEGFR2 signaling pathway	72	9.00	−35.01	−31.51
GO:0097435	GO Biological Processes	Supramolecular fiber organization	76	9.50	−33.32	−29.88
R-HAS-6809371	Reactome Gene Sets	Formation of the cornified envelope	40	5.00	−30.75	−27.58
R-HSA-72766	Reactome Gene Sets	Translation	52	6.50	−27.18	−24.11
R-HSA-5653656	Reactome Gene Sets	Vesicle-mediated transport	76	9.50	−25.95	−23.02
GO:1903311	GO Biological Processes	Regulation of mRNA metabolic process	47	5.88	−22.52	−19.70
GO:0006457	GO Biological Processes	Protein folding	38	4.75	−20.58	−17.87
hsa05132	KEGG Pathway	Salmonella infection	41	5.12	−20.15	−17.51
GO:0043254	GO Biological Processes	Regulation of protein-containing complex assembly	50	6.25	−19.00	−16.43
GO:0043086	GO Biological Processes	Negative regulation of catalytic activity	71	8.88	−18.67	−16.11
GO:0060341	GO Biological Processes	Regulation of cellular localization	70	8.75	−18.02	−15.51
GO:0051129	GO Biological Processes	Negative regulation of cellular component organization	64	8.00	−17.61	−15.16
R-HAS-75153	Reactome Gene Sets	Apoptotic execution phase	19	2.38	−16.53	−14.20
R-HSA-3371497	Reactome Gene Sets	HSP90 chaperone cycle for steroid hormone receptors (SHR) in the presence of ligand	19	2.38	−16.00	−13.72
GO:0022613	GO Biological Processes	Ribonucleoprotein complex biogenesis	48	6.00	−15.74	−13.47
Hsa04810	KEGG Pathway	Regulation of actin cytoskeleton	33	4.12	−15.21	−13.00
M71	Canonical Pathway	PID ILK Pathway	17	2.12	−15.14	−12.95

## Data Availability

The data presented in this study are available upon request.

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
