# Peer review of "Putative Wound Healing Induction Functions of Exosomes Isolated from IMMUNEPOTENT CRP"

_ijms, 2023, doi:10.3390/ijms24108971_

Round 1
Reviewer 1 Report
1. The authors used AFM to determine the size and shape for the exosome, however, the common method is TEM and NTA, could you explain the reason?
2. I see that there are already detailed data on data simulation, what is the author's next plan? Will more in vitro and in vivo biological experiments be done to verify it?
Author Response
We appreciate all the feedback we received from the reviewers. We hope to answer all of these satisfactorily. We will present each of the points mentioned by the reviewers and their corresponding answer and explanation.
Review report (Round 1):
- The authors used AFM to determine the size and shape for the exosome, however, the common method is TEM and NTA, could you explain the reason?
AFM was used in this study since it was previously employed by one of the co-authors in the study “Presence of Circulating miR-145, miR-155, and miR-382 in Exosomes Isolated from Serum of Breast Cancer Patients and Healthy Donors” to describe exosomes morphology if well TEM is the most common methodology exosome size can also be determined by AFM. As well as the fact that it is the method that we have available in our laboratory.
- Gonzalez-Villasana, V., Rashed, M. H., Gonzalez-Cantú, Y., Bayraktar, R., Menchaca-Arredondo, J. L., Vazquez-Guillen, J. M., Rodriguez-Padilla, C., Lopez-Berestein, G., & Resendez-Perez, D. (2019). Presence of circulating mir-145, Mir-155, and mir-382 in exosomes isolated from serum of breast cancer patients and healthy donors. Disease Markers, 2019, 1–9. https://doi.org/10.1155/2019/6852917
- I see that there are already detailed data on data simulation, what is the author's next plan? Will more in vitro and in vivo biological experiments be done to verify it?
The purpose of this manuscript is to report only the determinations of the in silico simulations of putative functions of the extract and the exosomes isolated from it. You're right, our team plans to report the findings from in vitro and in vivo experiments next in future investigations to verify our predictions, it would be important to include these insights in the conclusion, so we'll do that. We added the following in conclusion section:
In conclusion, we determine that the different biological activities of spleen cells dialyzed extract (IMMUNEPOTENT CRP) in part can be attributed to its total peptide content and peptides contained in exosomes, and not exclusive of a particular peptide. Furthermore, the in-silico analysis coordinates with activities described before such as anti-inflammatory capacity but a new activity such as wound healing process was founded, that permits to address of new investigation efforts toward this preclinical area to elucidate its activity, both the effect of the total ICRP extract and its exosomes. Predictions indicated a putative route in which the ICRP could exert a wound healing process through inflammation modulation and activation of signaling pathways such as PIP3-AKT among others related.
Reviewer 2 Report
Based on all the cited references, the main purpose of this paper is to determine putative functions of the ICRP through the use of bioinformatic tools based on the proteomic content of the extract and exosomes isolated from it in order to propose biological functions induced by the product. I think that although all the cited references suggested in the introduction are relevant to the research, originality / novelty not is displayed in the comparison with the cited references. Also, the introduction of study don’t provide sufficient background. You need to revise in that point.
Exosomes were isolated through ultracentrifugation, shape-size characterized by atomic force microscopy, and the protein content in the IMMUNEPOTENT CRP was obtained by EV-trap coupled to liquid chromatography. You explain how to do this in Chapter 4 by organizing sub-chapters from 4.1 to 4.4. In the case of this study, because materials and methods are important points, you need to explain them in detail in Chapter 4.
Most of this paper is composed of results. Of course, the results are important, but the process for deriving the results is as important as the results, so it is necessary to reduce the content of references cited in the results section and pay more attention to describing materials and methods. Also, I think that the conclusions are not supported by the results.
Author Response
Review report (Round 2):
Based on all the cited references, the main purpose of this paper is to determine putative functions of the ICRP through the use of bioinformatic tools based on the proteomic content of the extract and exosomes isolated from it in order to propose biological functions induced by the product. I think that although all the cited references suggested in the introduction are relevant to the research, originality / novelty not is displayed in the comparison with the cited references. Also, the introduction of study doesn’t provide sufficient background. You need to revise in that point.
The novelty of the study lies in the fact that the description of the content of the extract and the exosomes isolated from it, has never been reported before, nor had the presence of exosomes in the ICRP extract, or their possible biological functions based on protein content. We believe that the use of bioinformatics tools would be of great help to elucidate the possible mechanisms of this product. Although there is no novelty in the methodology used because it is only a tool that we use to propose possible functions, the novelty lies in these analyzes outside the already reported functions of the product. Since we only have background of biological functions in vitro, in vivo and at clinical trials of the ICRP related to cytotoxicity in cancer lines, anti-inflammatory and antioxidant effect, etc. but not of the possible functions of exosomes. A clarification was made in introduction:
Splenic stem cells of Hox11 lineage can differentiate when to injured exist in tissue into pancreatic islets [13] for that the spleen is now also considered as a source for cellular replacement, which could ameliorate b-islet cells. Suggesting the use of ICRP on diabetic wound.
We used a spleen dialyzed extract, no spleen cells. In addition to the suspect of the exosomes presence.
This modification was added in the introduction:
Nonetheless, based on the diversity of biological functions of ICRP, efforts have been done in the identification of compounds responsible for each function without relevant exit, concluding that ICRP activity when used as a complete spleen dialyzed extract of its biological activity is better in antitumoral activity. However new biological applications can be elucidated in a better manner when their peptides are identified and analyzed. Furthermore, previous studies using different enzymatic treatments on the whole extract do not affect the antitumor and antioxidant activities in vitro [14], and high temperature (90 ºC) for its production [15] raising suspicion that its activity could be attributed to the presence of exosomes, by this we centered in analyze if this questioning is correct. The main purpose of this paper is to determine putative functions of the ICRP using bioinformatics tools based on the proteomic content of the extract and exosomes isolated from it to propose biological functions induced by the product, that could indicate the pathway of the future investigations for diseases involving immunomodulation
Exosomes were isolated through ultracentrifugation, shape-size characterized by atomic force microscopy, and the protein content in the IMMUNEPOTENT CRP was obtained by EV-trap coupled to liquid chromatography. You explain how to do this in Chapter 4 by organizing sub-chapters from 4.1 to 4.4. In the case of this study, because materials and methods are important points, you need to explain them in detail in Chapter 4.
We hoped that modifications made in materials and methods section detail more the process followed.
4.1 IMMMUNEPOTENT CRP protein characterization
IMMUNEPOTENT CRP is a register product composed of a mixture of substances of <12 kDa from spleen extract manufactured by LONGEVEDEN SA de CV sold in a lyophilized presentation in Mexico. For proteomic analysis, a pool of lyophilized was created coming from different batches to assure homology and representative data of the product. Then this pool was analyzed by Tymora Analytics Operations (West Lafayette, IN) to identify the proteins, pool was digested to prepare a sample for liquid chromatography coupled to mass spectrometry coupled in tandem (LC-MS/MS) as described by Wu et al [36]. Company sends us the list of proteins present in the extract, that for bioinformatic analysis was considered as the target list.
4.1. Exosome proteomics characterization
From the pool of ICRP sent to Tymora Analytical Operations Company (West Lafayette, IN) a part of this mixture was separated to isolate the exosomes through EVtrap methodology with automated magnetic bead separation. After that lysed to liberate protein content. Then proteins were extracted and digested in an SDC buffer, later proteins were identified through LC-MS/MS analysis as specified by Wu et al [36]. This list of proteins was considered as target list for bioinformatics analysis. Since background and target lists were available, the company provided the relative abundance (%) of each protein on the target list compared to the background list.
4.2. Exosome shape and size characterization through atomic force microscopy
Exosomes were isolated from a pool from different batches of lyophilized ICRP employing the kit Exoquick (Invitrogen). That separates through ultracentrifugation extracellular vesicles. After that, the pellet was resuspended in sterile PBS 1X and analyzed by atomic force microscopy. The samples were observed using an NT-MDT Spectrum, NTEGRA Prima AFM at room temperature, with an RTESPA probe (Bruker) of spring constant k = 40 N/m in intermittent contact mode. Images of height, deflection, and phase were obtained; 20 x 20, 10 x 10, and 5 x 5 µm2 image sizes were captured systematically for each sample at three different regions at least. They were analyzed with Gwyddion software to observe the morphological aspect of exosome [16].
4.3. Biological pathway functions prediction
The background and target lists were composed of the gene names that codified the proteins. First, putative biological pathway functions were predicted on the online software Gene Ontology enRIchment analysis and visuaLizAtion tool (GOrilla) using a background list and target list (selecting as prediction models Homo sapiens and Mus musculus). Two simulations were performed in target list, one including all proteins present in the exosomes (1% to 100% relative abundance) and second one with overexpressed proteins in exosomes (6.06% to 100% relative abundance) with respect to the background list; for predictions employing the target list, the background list was introduced as reference. Same procedure was performed in Panther ontology specifying as model organism Bos indicus, Homo sapiens, and Mus musculus. Those software’s gave us data for molecular functions, cellular components, and biological processes.
For biological pathway prediction, proteins relation with transcription factors, and tissue specificity in Homo sapiens model the Metascape software was used. In the software, the two lists function was used to introduce background and target lists with automatic parameters.
4.4. Evaluation of wound healing related signaling pathways
Lastly to identify the proteins (interactors) that interacted with components of cell signaling pathways related to wound healing in literature and that correlated with the results obtained in previous softwares the Reactome online software was used entering background and target lists in analyze gene list section (enabling the button to show interactors in the results), with the PADOG analysis method, and introducing in species comparison sections “compare Homo sapiens with Bos taurus”. The program shows yellow coloration in components of the pathway that show interactions with the proteins in the lists. With this information, Supplementary tables 5 to 8 were built. Per signaling component interactor present in exosomes and ICRP were identified with their score of probability according to the Benjamini-Hocheber logarithm for multiple tests counting analysis.
- Gazze, S. A., Thomas, S. J., Garcia-Parra, J., James, D. W., Rees, P., Marsh-Durban, V., Corteling, R., Gonzalez, D., Conlan, R. S., & Francis, L. W. (2021). High content, quantitative AFM analysis of the scalable biomechanical properties of extracellular vesicles. Nanoscale, 13(12), 6129–6141. https://doi.org/10.1039/d0nr09235e
- Wu, X., Li, L., Iliuk, A., & Tao, A. (2018). Highly Efficient Phosphoproteome Capture and Analysis from Urinary Extracellular Vesicles. Journal of Proteome Research, 170(9), 3308-3316. https://doi.org/10.1021/acs.jproteome.8b00459
Most of this paper is composed of results. Of course, the results are important, but the process for deriving the results is as important as the results, so it is necessary to reduce the content of references cited in the results section and pay more attention to describing materials and methods. Also, I think that the conclusions are not supported by the results.
When we uploaded the article, an error was made and apparently the file of supplementary results was not attached, there more results are exposed that support the possible pathways that are being activated with the exosomes, these related to healing processes. In figure 14 can be observed that proteins present in the exosomes are interacting with several components of the PIP3-AKT pathway, highlighting charts in yellow representing an induction of the pathway. Also the following modification in conclusion section was made.
In conclusion, we determine that the different biological activities of spleen cells dialyzed extract (IMMUNEPOTENT CRP) in part can be attributed to its total peptide content and peptides contained in exosomes, and not exclusive of a particular peptide. Furthermore, the in-silico analysis coordinates with activities described before such as anti-inflammatory capacity but a new activity such as wound healing process was founded, that permits to address of new investigation efforts toward this preclinical area to elucidate its activity, both the effect of the total ICRP extract and its exosomes. Predictions indicated a putative route in which the ICRP could exert a wound healing process through inflammation modulation and activation of signaling pathways such as PIP3-AKT among others related.